# Clinical Determinants Affecting Indications for Surgery and Chemotherapy in Recurrent Ovarian Granulosa Cell Tumor

**DOI:** 10.3390/healthcare7040145

**Published:** 2019-11-14

**Authors:** Hidekatsu Nakai, Eiji Koike, Kosuke Murakami, Hisamitsu Takaya, Yasushi Kotani, Rika Nakai, Ayako Suzuki, Masato Aoki, Noriomi Matsumura, Masaki Mandai

**Affiliations:** 1Department of Obstetrics and Gynecology, Faculty of Medicine, Kindai University, Osaka 589-8511, Japan; e_koike@koike-byoin.com (E.K.); kmurakami@med.kindai.ac.jp (K.M.); htakaya@med.kindai.ac.jp (H.T.); y-kotani@med.kindai.ac.jp (Y.K.); guadangsuruya@yahoo.co.jp (R.N.); flapper@med.kindai.ac.jp (A.S.); aoki@med.kindai.ac.jp (M.A.); noriomi@med.kindai.ac.jp (N.M.); 2Department of Obstetrics and Gynecology, Kyoto Graduate School of Medicine, Kyoto University, Kyoto 606-8507, Japan; mandai@kuhp.kyoto-u.ac.jp

**Keywords:** recurrent granulose cell tumor, predictive factor, treatment strategy, surgery, adjuvant chemotherapy

## Abstract

Background: Because reports on the management of recurrent granulosa cell tumor have been sparse, a consensus as to which patients should undergo surgical resection and which patients should be considered for chemotherapy has not been established. Methods: A total of 21 tumor recurrences in eight patients with granulosa cell tumor were reviewed. Results: Surgery was performed as the main treatment for 13 recurrences, while chemotherapy was chosen as the main treatment for eight recurrences. Complete tumor resection could be accomplished in 13 of 16 surgeries (81.3%), which include all the ten recurrences without involvement of liver or diaphragm and without ascites. The number of recurrent masses was significantly higher in the early recurrence group (progression free survival < 2 years) than in the late recurrence (progression free survival > 2 years). All cases with a solitary recurrent tumor at an extra-peritoneal site presented a significantly longer progression free survival. Conclusions: For patients with recurrent granulosa cell tumor, surgery may provide the best disease control. In cases with complete resection, the number of recurrent masses was the predictive factor for the next recurrence, and adjuvant chemotherapy might be considered in such cases.

## 1. Introduction

Granulosa cell tumor (GCT) is a rare ovarian malignancy which accounts for 2% of all ovarian malignant tumors [1]. Patients with this tumor are generally regarded as having a good prognosis. Partly because more than 90% of GCT is diagnosed at stage I, the 5-year overall survival (OS) rate of GCT is 90–95%. However, 20–25% of GCT ultimately recurs over a long period [2,3]. Furthermore, recurrent GCT may recur multiple times, and 70–80% of the patients with recurrent GCT eventually die of the disease [4]. Therefore, the long-term prognosis of GCT is not as encouraging as its short-term survival: the 10-year and 25-year OS are 60–90% and 40–60%, respectively [5,6]. In this regard, the management of recurrent GCT is an important health issue as well as the treatment of primary disease.

The management of primary GCT has been the topic of many reports. The recommended surgical procedure, indication for adjuvant chemotherapy, and choice of chemotherapeutic agent have been described in the national comprehensive cancer network (NCCN) guideline [7]. However, in this guideline, the accepted treatments for recurrent GCT include only salvage chemotherapy, anti- vascular endothelial growth factor (VEGF) therapy, radiotherapy, and hormone therapy. Although VEGF is expressed in GCT [8], only limited efficacy (6 PR/36 cases) was observed in a clinical trial [9]. In the literature, aggressive surgery for recurrent GCT has been reported to improve the prognosis, and the rate of success of optimal surgery in recurrent GCT is assumed to be as high as 80–85% [10]. Because reports on the management of recurrent GCT have been sparse, a consensus as to which patients should undergo surgical resection and which patients should be considered for chemotherapy has not been established.

Regarding the chemotherapy regimen for recurrent GCT, Bleomycin Etoposide Cisplatin (BEP) has been recommended based on the GOG trial [11]. According to the NCCN guideline, taxane +/− carboplatin is also recommended for recurrent GCT. Considering that treatment for recurrent GCT often consists of multiple surgeries with multiple organ resections, a highly toxic regimen such as BEP is not ideal in terms of patient quality of life and feasibility of treatment. However, reports on other chemotherapeutic drugs for recurrent GCT have been rare. In addition, the indication for adjuvant chemotherapy after surgery for recurrent GCT is unknown.

In this study, we retrospectively reviewed 21 treatments for recurrent GCT in an attempt to find predictive factors for successful surgery among the indications for surgery. In addition, we also sought to identify prognostic factors after surgery to assist in determining the indication for post-operative adjuvant chemotherapy.

## 2. Materials and Methods

A total of 21 tumor recurrences in eight patients with ovarian GCT treated at Kindai University Hospital between 1996 and 2010 were included in this study. As the primary treatment, unilateral salpingo-oophorectomy (USO) was performed at a minimum and the pathological diagnosis was confirmed by a gynecologic pathologist. Among eight patients, seven (87.5%) had repeated recurrences, with a total number of 21 recurrent events in these. Surgery was performed as the main treatment for 13 recurrences (surgery-group), while chemotherapy was chosen as the main treatment modality in the remaining eight recurrences (CT-group). In three of these recurrences, laparotomy was actually performed, but only tissue biopsy was sampled due to the presence of unresectable tumor; these cases were analyzed as part of the CT-group. In 16 recurrences (13 in the surgery-group plus three in the CT-group), the diagnosis of recurrent GCT was confirmed pathologically, while the diagnosis of recurrence was made only by computed tomography in the other five recurrences. In each recurrence, the patient characteristics, method of treatment, and progression-free survival (PFS) after treatment were analyzed.

Participants were fully informed of procedures, and written informed consent was obtained from each. The ethics committee for human studies at Kindai University faculty of medicine approved the present study, which was conducted in accordance with the Declaration of Helsinki (Ethics board approval number is 27-081).

Comparison of the categorical data was performed using the Χ^2^ and t tests for unpaired data. The Kaplan–Meier curve was determined by log-rank analysis of progression-free survival. All tests were two-tailed, and the results were deemed to be statistically significant at a level ≤0.05.

## 3. Results

The clinical characteristics of the eight patients are shown in Table 1.

The median age of the patients was 44 years (range 32–68). For their primary lesion, seven patients were diagnosed as stage Ic, and one patient as stage IIb. As their primary surgery, five cases underwent USO, one case underwent abdominal total hysterectomy and bilateral salpingo-oophorectomy (ATH + BSO), and two cases underwent ATH + BSO with pelvic lymph-node dissection. None of them received adjuvant chemotherapy. The median recurrence-free survival was 56.5 months (range 14–168) and the median follow-up period was 135 months (range 101–349).

The treatment course of the eight cases are shown in Figure 1.

Recurrences were divided into three groups; (i) recurrence at abdominal cavity without liver/diaphragm involvement and without ascites, (ii) involvement (including surface) of liver/diaphragm with or without ascites, (iii) peritoneal dissemination with ascites. Surgeries were performed for all the ten recurrences without liver/diaphragm and without ascites, all of which achieved macrospically complete resection, followed by adjuvant chemotherapy in only one case (case 1). Of the eight recurrences involving the liver and/or diaphragm, chemotherapy was selected for four cases, none of whom achieved a complete response, and surgeries were performed for three cases, which included one complete (case 2) and two incomplete surgeries (case 3). One patient did not wish to receive anti-tumor treatment (case 7). Of the four recurrences with peritoneal disseminations associated with ascites, pre-operative chemotherapy was performed for two cases followed by complete surgeries (case 5 and case 7). One case received surgery followed by chemotherapy (case 8) and another case received only chemotherapy (case 4). The number of patients who survived without disease was four, three, two, and one at five, ten, twenty, and thirty years after the initial treatment, respectively. Regarding the intervals to the next recurrence, the median progression free interval from the treatment of the first recurrence to the second recurrence, second to third, and third to fourth was 21, 27, and 19 months, respectively, with no significant difference among them (Figure 2).

Factors predictive of early recurrence were analyzed by comparing early and late recurrences (Table 2).

Early recurrence was defined as a recurrence that occurred within 2 years after the previous recurrence (PFS < 2 years group) and late recurrence as that occurred more than 2 years after the previous recurrence (PFS > 2 years group). There was no significant difference between these two groups with regard to the site of recurrence, but in three recurrences in the PFS > 2 years group, the tumor recurrence involved the inferior rectus and oblique abdominal muscles (extra-peritoneal localization), and comparatively long-term PFS were observed (25, 66, and 67 months). In those three recurrences, the median PFS was significantly longer than the other 10 surgical treatment cases (*p* = 0.013). In addition, the mean number of tumors was significantly greater (8.4 ± 3.8, range 3–16) in the PFS < 2 years group than in the PFS > 2 years group (4.8 ± 2.1, range 3–10) (*p* = 0.03). There was no significant difference in terms of mean size of tumor, mean number of previous recurrences, tumor rupture and positive peritoneal cytology, and use of adjuvant chemotherapy.

The efficacy of chemotherapy in the CT-group is shown in Table 3.

Because of the presence of cancerous ascites and pleural effusion in five cases, these patients had a poor performance status. In one case, the patient had bone marrow suppression due to previous chemotherapy for breast cancer. For this reason, BEP therapy was indicated in only one patient. The other regimens of chemotherapy included TC (paclitaxel + carboplatin) in three treatment courses, gemcitabine in two, cyclophosphamide + cisplatin in one, and cyclophosphamide in one. The mean number of chemotherapy courses was 9.9 ± 7.9 (range 2–26). The overall response according to the revised response evaluation criteria in solid tumors (RECIST) guideline ver. 1.1 was 62.5% (5/8), and a complete response was observed in 25% (2/8). The median PFS was 38 months and the median overall survival was 48 months. Adverse events according to CTC-AE ver. 4.0 are shown in Table 4.

A good recovery was noted for all the adverse events and there was no treatment-related death.

## 4. Discussion

Management for primary GCT in terms of method of surgery, efficacy of adjuvant chemotherapy, and adjuvant radiotherapy has largely been standardized by the previous studies [12,13,14,15,16]. Incomplete surgical staging was associated with increased hazard of death in primary GCT [17]. The response rates for chemotherapy using BEP or TC were 37–90% [11,15,18], and those for hormonal therapy using aromatase inhibitor, leuprorelin, or tamoxifen were 40–71% [19,20]. On the other hand, management for recurrent GCT is not established yet. Problems to be solved include indication of surgery, predictive factor for next recurrence, and efficacy of post-operative adjuvant chemotherapy.

Whether surgical resection of recurrent GCT is possible or effective was first evaluated. In this study, the number of recurrent tumors per case was greater than three in all treated cases. However, complete resection could be accomplished in 13 of 16 surgically treated cases (81.3%), including a case with 16 recurrent masses. The size of tumor, mean number of recurrent masses, mean number of recurrences, and mean interval from the previous recurrence did not influence the choice of surgery. Likewise, in the literature, more than half of recurrent GCT has been reported to occur at multiple sites. The optimal debulking rate has been assumed to be as much as 80–85%, and optimal debulking is considered to improve prognosis [10,21,22]. It has been suggested that optimal surgery should be considered in most cases regardless of the number of recurrent tumors, unless ascites and pleural effusion are observed. We considered a case with recurrence close to porta hepatis and peritoneal carcinomatosis to be inoperable, but Chua et al. reported that reduction with radical peritonectomy was possible in recurrent GCT [23]. The median disease-free survival of five such cases was 10–95 months, although the observation period was short (median follow-up period was 38 months). There were relatively frequent peri-operative complications of grade 3/4. Some case reports suggest the usefulness of aggressive surgery, including resection of hepatic metastasis [24,25]. The indication for aggressive surgery needs to be further evaluated.

In the case of epithelial ovarian cancers, the intervals to second and third time of recurrence tend to be shorter than the previous ones and the treatment becomes more difficult with repeated recurrence. By contrast, in our study, there was no significant difference in the intervals as recurrences occurred. Fotopoulou et al. reported in 27 recurrent GCT cases that the median PFS from the first to the second recurrence was 20 months, while that from the second to the third recurrence was 18 months [26]. In the MITO-9 trial, which included 35 recurrent GCTs, the median PFS from the first to the second recurrence was 38 months and that from the second to the third recurrence was 41 months [27]. Several other reports with a smaller number of cases yielded similar results. These data suggest that an important characteristic of GCT, namely, the intervals between repeated recurrences may be maintained overtime, which may boost the efficacy of surgery.

Another important issue in the management of recurrent GCT is the use of adjuvant therapy after surgery, about which very few reports have appeared. The MITO-9 trial did not recommend post-operative adjuvant chemotherapy for recurrent GCT because 11 of 35 (31.4%) cases recurred despite post-operative adjuvant therapy being performed in 81.8% of all cases. However, in most of the previous studies, as much as 70–80% of GCT recurrences recurred again and most of these patients died of disease [4,28]. By comparison, the recurrence rate in the MITO-9 trial appears to be less than that reported here and in other reports. In the MITO-9 trial, second and third recurrences occurred only in 11 (31.4%) and four cases (36.4%), respectively. The median PFI between the first to second recurrences and second to third recurrences were 38 and 41 months, respectively, which were apparently longer than reported in other studies. These differences may have resulted from a higher rate of adjuvant chemotherapy in the MITO-9 trial. At a minimum, it appears that the intervals between the recurrences tended to become longer with adjuvant therapy in primary GCT [29]. However, it is still unclear whether the administration of adjuvant therapy contributes to the final recurrence rate, as a later recurrence could occur in these cases.

To determine which case actually would be the indication of adjuvant therapy, we investigated predictive factors for recurrence after surgery in recurrent GCT. We compared the patient characteristics between cases with late recurrence after previous treatment (PFS > 2 years) and those with early recurrence (PFS < 2 years). No significant difference could be found in the site of recurrence, the size of tumor, the number of recurrences, tumor rupture, and positive peritoneal cytology. In contrast, the number of recurrent masses was significantly higher in the early recurrence group, suggesting the residual presence of micro metastatic foci in the peritoneal cavity. Therefore, post-operative adjuvant therapy may be considered particularly in cases with multiple intra-peritoneal recurrences. In contrast, all cases with a solitary recurrent tumor at an extra-peritoneal site presented a significantly longer PFS. These recurrences may represent implantation in a surgical wound, which has been occasionally noted in GCT [30]. In these cases, adjuvant chemotherapy could be avoided.

In this study, various chemotherapy regimens were used, including BEP, TC, CP, cyclophosphamide, and gemcitabine. The standard chemotherapy for recurrent GCT is BEP therapy based on the phase II GOG trial [9] and PVB therapy based on the EORTC trial [31]. In spite of a high response rate to these regimens, the median PFS are not prolonged, and severe adverse effects have been noted. For less toxic regimens, the clinical usefulness of TC, CP, and cyclophosphamide for recurrent GCT has been recently reported [13,15,32,33]. However, there is no report on the use of gemcitabine for recurrent GCT. In our series, only a few cases achieved CR, although relatively long-term disease control was achieved with chemotherapy with mild toxicity. Considering that a median PFS of 19 months was obtained by tumor resection alone, we should primarily consider surgical resection for recurrent GCT when possible.

## 5. Conclusions

In summary, for patients with recurrent GCT, surgery may provide the best disease control compared with other treatments, and optimal surgery could be accomplished in most of the cases. In cases with complete resection, the number of recurrent masses was the predictive factor for the next recurrence, and adjuvant chemotherapy might be considered in such cases. With regard to chemotherapy regimens, the less toxic regimens including gemcitabine may be considered because long-term management is necessary in many cases of recurrent GCT due to repeated recurrence. Because a randomized trial may not be possible for this rare disease, the continued accumulation of treatment experiences is needed.

## Figures and Tables

**Figure 1 healthcare-07-00145-f001:**
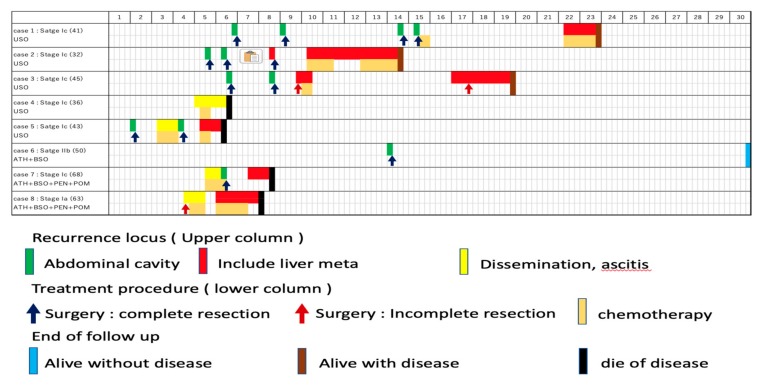
Treatment course of the eight cases. The first row shows case number, stage, age, and the initial surgery. The first column shows years following the initial surgery. Recurrence locus (upper column), treatment procedure (lower column), and end of follow up are shown as indicated.

**Figure 2 healthcare-07-00145-f002:**
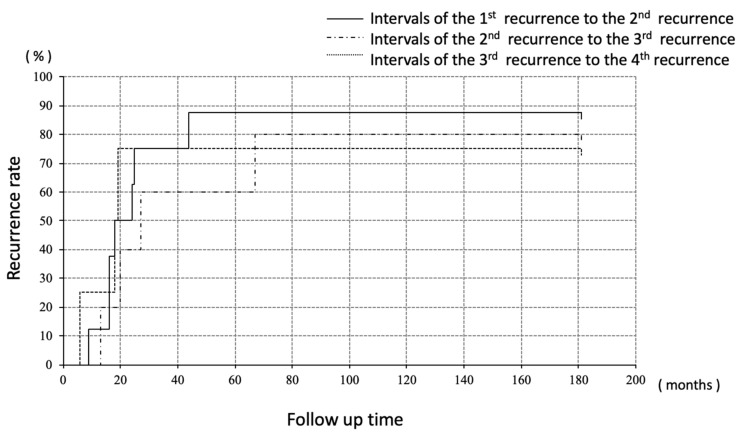
Median PFI every recurrence treatment; no significant deference was seen.

**Table 1 healthcare-07-00145-t001:** Patients’ background.

Patients’ Background	
Number of cases	8
Median age (range)	44 (32–68)
Primary surgery	
USO	5
ATH + BSO	1
ATH + BSO + PEN + POM	2
Stage	
Ιa	1
Ιc	6
Ⅱb	1
Menopausal status	
Menopause	2
Premenopause	6
Median recurrence free period (range, month)	56.5 (14–168)
Median follow up period (range, month)	135 (101–349)

USO: unilateral salpingo-oophorectomy; ATH: abdominal total hysterectomy; BSO: bilateral salpingo-oohphorectomy; PEN: pelvic lymph-node dissection; POM: partial omentectomy.

**Table 2 healthcare-07-00145-t002:** Prognostic factor of complete surgery.

Factor	PFS > 2 Years	PFS < 2 Years
Site		
extra-pelvis *	3	0
include upper abdomen	2	5
pelvis	1	2
Rupture		
+	4	2
−	2	4
Mean number of tumor (range) *	4.8 ± 2.1 (3–10)	8.4 ± 3.8 (3–16)
Mean size of tumor (range, cm)	3.0 ± 1.5 (1–7)	2.1 ± 1.0 (1–8)
Peritoneal washing		
Positive	2	2
Negative	4	5
Adjuvant chemotherapy (without oral)		
+	1	1
−	5	6

PFS: progression-free survival; * *p* < 0.05.

**Table 3 healthcare-07-00145-t003:** Efficacy of chemotherapy.

Regimen	Direct Effect	Progression Free Survival (Months)
Paclitaxel + Carboplatin (n = 3)	CR/CR/PR	12/41/84
Bleomytin + Etoposide + Cisplatin (n = 1)	PR	12
Cyclophosphamide + Cisplatin (n = 1)	SD	18
Cyclophosphamide (n = 1)	PD	0
Gemcitabine (n = 2)	SD/PD	30/0

**Table 4 healthcare-07-00145-t004:** Adverse effect of chemotherapy.

Adverse Event	Grade 3/4
Hematologic toxicity	
neutrocytopenia	6/8
anemia	2/8
thrombocytopenia	1/8
febrile neutropenia	0/8
Non-Hematologic toxicity	
gastro-intestinal	2/8
pulmonary fibrosis	1/8
neurotoxicity	0/8

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
