# Peer review of "Clinical Determinants Affecting Indications for Surgery and Chemotherapy in Recurrent Ovarian Granulosa Cell Tumor"

_healthcare, 2019, doi:10.3390/healthcare7040145_

Round 1
Reviewer 1 Report
A key reference supports the role of surgery in the National Cancer Database published by Seagle et al in Gynec Oncol 2017. This case series supports a role for surgery in recurrence. The presentation of the data in this revision has improved.
Specific suggestions: Materials and methods line 65 and 68 instead of repeated 8 patients, just substituted by 'these'. And continue In 3 of 'these' recurrences (rather than '8 latter'.
Table 3: indicate Progression Free Survival (months)
Reviewer 2 Report
Great work; Congratulations to the authors!
Accepted the current version.
Author Response
Thank you very much for your comments.
This manuscript is a resubmission of an earlier submission. The following is a list of the peer review reports and author responses from that submission.
Round 1
Reviewer 1 Report
This study evaluates clinical determinants affecting indications for surgery and chemotherapy in recurrent ovarian granulosa cell tumor. Definitely deserves to be published and is a valuable contribution to the “Healthcare” Journal. Some minor flaws could be addressed before publication.
Minor points:
[1] Introduction, lines 43-44: Please, make a comment about the available evidence for the common expression of vascular endothelial growth factor (VEGF) in granulosa cell tumors, and the relevant therapeutic role of bevacizumab that could be especially valued [Boussios S, et al, Ovarian sex-cord stromal tumours and small cell tumours: Pathological, genetic and management aspects. Crit Rev Oncol Hematol. 2017;120:43-51].
[2] Results: It could be interesting to incorporate in Table 1 the stage of the disease at the initial diagnosis of those 8 patients. This is the most important determinant of prognosis for granulosa cell tumors. You can also mention the initial menopausal status of the patients.
[3] Results: Please, correct the typo in Table 4; “gemcitabine” instead of “gemcitabin”.
[4] Discussion, lines 138-139: Which are the achieved response rates for patients treated with chemotherapy according to the literature? What about the reported efficacy of the hormonal treatment?
Reviewer 2 Report
This is a relatively small series of women with granulosa cell tumors that undergo treatment for recurrence.
Their analyses are sound but rendered unnecessarily complex by classifying the results in 8 patients into a surgery and a chemotherapy group (Table 2). While analyzing separately the results of these interventions, since we are dealing with the same patients often undergoing such procedures, intuitively it is more relevant to present the data as they apply to each of the eight patients the constitute the experience of this center. Tables 3 and 4 and figure 1 are still applicable as is the description in the abstract, except the statement of complete tumor recurrence 'accomplished in 13 of 16 surgically treated cases' doesn't make sense. Also comparing sites of recurrence in Surgery-group and absence of peritoneal carcinomatosis in the Surgery-group is not worth the emphasis in the abstract. Description and outcomes of those patients who undergo surgery should be the focus of this small series. The artificial comparison of the two groups should be de-emphasized.
In the chemotherapy group (table 4) etc should be listed as miscellaneous (see text) and not etc. The numbers are small enough to list with each of the 4 groupings one should indicate actual individual PR/CR/no response and individual PFS. There is no mention of hormonal therapy, and review(s) of its use should be cited, since aromatase inhibitors have recently received emphasis in view of the FOXL2 mutation diagnostic of this stromal ovarian tumor
